# Development of reporting guidance and core outcome sets for seamless, standardised evaluation of innovative surgical procedures and devices: a study protocol for content generation and a Delphi consensus process (COHESIVE study)

Kerry Avery,[1] Jane Blazeby,[1,2] Nicholas Wilson,[1] Rhiannon Macefield,[1] Sian Cousins,[1] Barry Main,[1,2] Natalie S Blencowe,[1,2] Jesmond Zahra,[1] Daisy Elliott,[1] Robert Hinchliffe,[1,3] Shelley Potter[1,4]

KA and SP contributed equally.

For numbered affiliations see end of article.

**Correspondence to**
Dr Kerry Avery;
kerry.avery@bristol.ac.uk

## ABSTRACT

**Introduction** Rigorous evaluation of innovative invasive procedures and medical devices is uncommon and lacks reporting standardisation. Devices may therefore enter routine practice without thorough evaluation, resulting in patient harm. Detailed guidance on how to select and report outcomes at each stage of evaluation is lacking. Development of reporting guidance and core outcome sets (COS) is one strategy to promote safe and transparent evaluation.

**Methods and analysis** A COS, comprising outcome domains applicable to all phases of evaluation of procedure/device introduction and modification and, if necessary, supplementary domains relevant to specific phases or types of innovation (procedure or device), will be developed according to principles outlined by Core Outcome Measures in Effectiveness Trials (COMET) and Core Outcome Set-Standards for Development (COS-STAD) guidelines. Reporting guidance will be developed concurrently. The study will have the following three phases:

1. Generation of a list of relevant outcome domains and reporting items identified from (a) published studies, (b) hospital policy documentation, (c) regulatory body documentation and (d) stakeholder qualitative interviews. Identified items/domains will be categorised using a conceptual framework and formatted into Delphi consensus survey questionnaire items.

2. Key stakeholders, including 50 patients and 150 professionals (surgeons, researchers, device manufacturers, regulatory representatives, journal editors) sampled from multinational sources, will complete a Delphi survey to score the importance of each reporting item and outcome.

3. A consensus meeting with key stakeholders will discuss and agree the final content of the reporting guidance and COS(s).

### Strengths and limitations of this study

► Consensus methods will be used to develop guidance and core outcome sets (COS) for seamless, standardised evaluation and reporting of outcomes of innovative invasive procedures and medical devices.

► Guidance and COS(s) will be developed with multinational, multi-stakeholder input, including patients, surgeons, device manufacturers, regulators, methodologists and journal editors.

► Multiple novel data sources beyond traditional systematic reviews will generate a list of guidance items and outcomes of relevance to evaluating innovative procedures and devices.

► An integrated randomised methodological study will investigate methods for optimally achieving consensus during Delphi surveys.

► Further work is needed to explore how to optimally implement and monitor use of the guidance and COS(s).

**Ethics and dissemination** Ethical approval has been granted by North East-Newcastle and North Tyneside 1 Health Research Authority Research Ethics Committee (18/NE/0378). Dissemination strategies include scientific meeting presentations, peer-reviewed journal publications, development of plain English summaries/materials, patient engagement events, development of a social media identity, workshops and other events.

## INTRODUCTION

Methods for introducing innovative invasive (surgical) procedures and medical devices (any devices used inside the body including

implantable surgical devices, eg, pacemakers, mesh, implants) into clinical practice are not optimally regulated. Rigorous early evaluation of procedures/devices is uncommon, and reporting of outcomes is neither comprehensive nor standardised.[1] Suboptimal regulation and reporting has resulted in compromised patient safety,[2–4] highlighting the need for robust, transparent methods for safe and efficient innovation.[1]

The IDEAL framework was published in 2009 to improve the evaluation of invasive surgical procedures from first in man use to long-term study.[5] More recently, IDEAL-D has been introduced to facilitate the evaluation of medical devices.[6] Adoption of IDEAL/IDEAL-D by the surgical community has been slow.[7] Reasons for this may be attributed to lack of understanding around applying the framework in practice.[7] Guidance on selecting, measuring and reporting outcomes relevant to evaluating invasive procedures/devices during specific stages of their introduction and modification is also lacking. Comprehensive evaluation and reporting would allow learning to be shared across the surgical community and enhance transparency, allowing promising innovations to be identified more rapidly and unsafe or harmful procedures/devices to be abandoned before they become established into routine clinical practice. Streamlined mechanisms for evaluating breakthrough technologies and treatments and speeding up their adoption is the primary objective of the UK government's Accelerated Access Review to improve patient outcomes.[8] Appropriate outcome selection is integral to these processes. Development of guidance and a generic core outcome set (COS: a minimum set of outcome domains to be reported in all uses in clinical practice, studies or audits) may be one strategy by which safe and efficient introduction and modification of novel procedures/devices can be achieved.

While methods for developing COS for effectiveness studies are well established, it is unclear whether these methods will be applicable to developing a COS for innovative invasive procedures/devices. The process of selecting outcomes for effectiveness studies, in which the intervention under evaluation is expected to be stable, is unlikely to identify outcomes of specific relevance to innovation, such as technique modification or abandonment. It is also important that some outcomes included within the COS are valid throughout all stages of evaluation to allow for earlier identification of longer-term adverse events. To achieve this, it is hypothesised that a modular COS approach may be needed to capture outcomes relevant to different stages of evaluation. This may include a central generic COS that would be measured and reported in all uses and evaluations, supplemented by bolt-on COSs relevant to specific phases of innovation and/or evaluation, and, if necessary, the specific disease or intervention. The necessity and framework for a modular COS approach will be evaluated throughout the study.

## Aim

The Core Outcomes for early pHasE Surgical Innovation and deVicEs (COHESIVE) study will develop reporting guidance and a COS(s) for seamless, standardised outcome evaluation of innovative procedures and devices.

## METHODS AND ANALYSIS
### Overview

The COS(s) will be developed according to the Core Outcome Measures for Effectiveness Trials (COMET) Handbook[9] and the Core Outcome Set-STAndards for Development (COS-STAD) guidelines.[10] Reporting guidance will be developed concurrently. The study will have three phases:

1. Generation of a comprehensive list of reporting guidance content (eg, items relating to guidance about additional detail required when reporting a specific outcome) and outcome domains from multiple data sources.
2. Prioritisation of identified items using a multistage, international stakeholder Delphi survey.
3. A stakeholder consensus meeting to discuss, agree and ratify the final guidance and COS(s).

Reporting guidance will be developed in line with existing methodologies,[11] including involvement of an executive group, Delphi consensus methods and a face-to-face meeting of stakeholders to maximise the robustness of guidance development.

### Patient and public involvement

Patients and members of the public will be involved throughout. A patient and public involvement (PPI) group has been established as part of the National Institute for Health Research (NIHR) Bristol Biomedical Research Centre (BRC) Surgical Innovation theme (of which this study is a part). Two patients have been appointed to the study steering group and will be invited to contribute to all study phases and discuss dissemination of the findings.

### Scope of the guidance and COS(s)

The reporting guidance and COS(s) will support standardised evaluation of innovative invasive procedures/devices. The COS(s) will apply to routine clinical practice, audit and research settings where introduction and modification of invasive procedures/devices occur prior to definitive evaluation within a pragmatic randomised controlled trial.[5] It is anticipated, however, that some outcomes will be 'core' to stages of evaluation beyond introduction and modification (eg, including registries and long-term studies) to allow key outcomes to be evaluated from the earliest studies of an invasive procedure/device. The guidance and COS(s) will be developed for use by key stakeholders involved in the introduction, modification, evaluation or reporting of innovative invasive procedures/devices, including surgeons, device manufacturers, regulators, trialists, methodologists and journal editors.

## Definition of an invasive procedure and medical device

An invasive procedure is defined as one where access is gained via an incision, natural orifice or percutaneous puncture or involving devices used inside the body.[12] An invasive medical device will be further defined as one that is inserted into and remains within the body with the aim of achieving a therapeutic function.

## Definition of an 'innovation' outcome

While work to develop a definition of surgical innovation is ongoing,[12] no single definition of surgical innovation has yet been widely adopted. The *COMET Handbook* defines an outcome as a measurement or observation used to capture and assess the effect of treatment such as assessment of side effects (risk) or effectiveness (benefits).[9] Although relevant for effectiveness trials, this definition is unlikely to capture all issues of relevance to evaluation of novel invasive procedures/devices.[13] Preliminary work by the study team indicates that, in addition to established effectiveness outcomes, additional outcomes may be required to fully evaluate the process and effects of innovation, such as modification or abandonment of the technique/device.[14] An innovation outcome will therefore be defined as a measurable variable (eg, construct or concept) related to or occurring as a result of the use of a novel invasive procedure/device. An outcome domain will be defined as a group of individual outcomes.[13]

## Stakeholder involvement

A workshop funded by the Medical Research Council Hubs for Trials Methodology Research Network was convened in Bristol, UK, in September 2018. The workshop brought together 61 key stakeholders, including surgeons, representatives from funding bodies, device industries and small and medium sized enterprises, trialists, methodologists, journal editors, regulators and patient representatives, to consider core outcome reporting. Stakeholders represented several countries, including the UK, USA, Canada and Australia. The workshop concluded that development of reporting guidance and a COS(s) for innovative surgical procedures and devices via methods described in this study would be feasible and warranted further study.

The guidance and COS(s) will be developed with input from multiple stakeholder groups at all stages, to ensure that outcomes of relevance to all are included and for the COS to be widely adopted.[9] Primary stakeholders, recruited from international sources, will include the following:

► Patients with experience of surgery and/or undergoing innovative surgical procedures.
► Surgeons (those involved/not in delivering innovative procedures/devices).
► Device manufacturers and members of industry.
► Representatives of regulatory bodies and policy makers.
► Journal editors.

► Methodologists and trialists responsible for designing studies involving innovative invasive procedures/devices.

A study steering group comprising stakeholder representatives from the UK and internationally will be convened to provide overall oversight of the study.

## Phase I: generation of a list of outcome and reporting domains

As innovation outcomes are conceptually different from those measured in effectiveness trials, standard methods[9] for identifying outcomes for inclusion in the list will be modified to include data sources hypothesised to be of specific relevance to innovation.

### Targeted review of outcome selection and reporting recommendations in international regulatory body documentation

Online searches will identify advisory documents from prominent regulatory bodies of relevance to innovation of invasive procedures/devices, including the US Food and Drug Administration, UK Medicines and Healthcare products Regulatory Agency, UK National Institute for Health and Care Excellence Interventional Procedures Advisory Committee, EU Notified Bodies and Conformité Européene (CE) marking guidance to explore existing recommendations regarding outcome selection and reporting. Documents will be reviewed, and data relating to outcome selection and reporting extracted verbatim.

### Focused literature reviews of outcome selection and reporting in published studies of invasive procedures/devices

A series of focused literature reviews will explore outcome selection and reporting in studies of innovative invasive procedure/devices.

#### Case studies of innovative devices

Case studies of innovative devices will be purposively selected by the study team to include a range of specialties and varying degrees of innovation (wholly innovative—defined as a device that represents a completely new approach to solving a clinical problem; partially innovative—defined as a device that is broadly similar in function to one already in use but differs in at least one significant way; reinvented—defined as a modification of a device or technique that was previously abandoned for complications). A focused PubMed search will be performed using the tradename of each identified device and reference lists of identified papers, reviews and commentaries to identify potentially relevant studies. Only primary studies specifically related to the device under evaluation will be included. Data regarding study design and timing of evaluation in the device lifecycle (eg, pre-market vs post-market) will be extracted. Outcomes selected and reported in each study will be extracted verbatim, and pre-marketing and post-marketing studies compared to explore whether outcome selection and reporting differs over time and to inform development of the reporting guidance and COS(s).

### Studies conducted within the IDEAL and IDEAL-D frameworks

It is hypothesised that studies self-identifying as IDEAL[5] or IDEAL-D[6] (ie, where authors self-report that the study aligns with a stage of the IDEAL/IDEAL-D framework) may have more formally considered outcome selection in the context of innovation and may therefore provide greater insight into potential 'innovation' outcomes. Studies citing the three original 2009 Lancet publications,[5 15 16] subsequent explanatory documents[17–22] or the IDEAL-D paper[6] will be identified through the Web of Science and Scopus databases. All primary papers self-identifying as IDEAL/IDEAL-D studies in the title or abstract will be included. Outcomes selected and reported in each study will be extracted verbatim and, if possible, compared across IDEAL stages (1, 2a, 2b) to explore if different outcomes are evaluated at different stages of innovation and to inform development of the reporting guidance and COS(s).

### Review of recommendations for outcome selection and reporting in NHS trust policy documents

Written NHS Trust policy documents for the introduction of new invasive procedures/devices into clinical practice will be reviewed to explore current requirements for outcome selection and reporting within the NHS governance framework. Documents will be identified from related work ongoing in the NIHR Bristol BRC Surgical Innovation theme. Documents will be requested from all acute NHS Trusts in England and Health Boards in Wales. Initially, purposive sampling will be used to sample Trusts based on type, geographical location and foundation status. All outcomes relating to the evaluation of innovative devices will be extracted verbatim. Purposive sampling will continue until new outcomes cease to be identified (data saturation).

### Review of transcripts of qualitative interviews with patients and surgeons

Transcripts of audio-recordings from a purposive sample of qualitative interviews with key stakeholders (eg, patients, surgeons and industry representatives) will be reviewed to identify additional outcomes of relevance that have not been identified from other data sources. These interviews are being conducted for related work undertaken in the NIHR Bristol BRC Surgical Innovation theme.

### Conceptualisation of outcome domains and reporting items

Verbatim outcomes identified from each data source will be used to create a comprehensive list of potential outcomes relevant to the evaluation of novel invasive procedures/devices. Preliminary work by the study group has demonstrated the importance of context (eg, specialty and innovation stage) to outcome classification for invasive procedure/device studies. A conceptual framework of outcome domains will therefore be generated and iteratively modified using each case study and data source in turn until saturation is reached (ie, no new outcomes emerge). Conceptualisation will be undertaken independently by members of the study team, followed by a meeting to review, discuss and modify the framework. Conceptualisation, review and refinement will be undertaken iteratively until data saturation is achieved and the list is considered complete.

Content relevant to inform development of the reporting guidance will be accumulated and iteratively refined through ongoing discussion within the team. Guidance content and formatting will be informed by a targeted review of relevant research guidance documents (eg, IDEAL, CONsolidated Standards Of Reporting Trials/CONSORT statement and CONSORT extensions).

### Phase II: prioritisation of identified reporting items and outcome domains using a multi-stakeholder Delphi survey

A consensus process involving a sequential, multi-round Delphi survey followed by a face-to-face consensus meeting will be used to establish agreement between a multi-stakeholder group of patients and professionals on the final set of outcome domains to be included in the COS(s). To avoid any effect of dominant individuals, a Delphi survey consensus process will allow a diverse representative sample of key stakeholders from a broad geographical area to participate anonymously.[9]

### Development of the Delphi survey questionnaire

Each outcome domain included in the final outcomes list detailed above will be operationalised and formatted into an item for the survey questionnaire. Each item will have a 9-point Likert scoring scale ranging from 1 (not important) to 9 (extremely important), based on the Grading of Recommendations Assessment, Development and Evaluation scale for scoring the importance of including the item in the final COS.[23] A free text item will be included to enable participants to propose new items. Additional outcome domains/items proposed by participants in round 1 will be reviewed by the study team to confirm that they are new and formatted into a survey item for subsequent round(s) if they are recommended by at least two participants.[24] Any uncertainties will be resolved by the study steering group. To ensure that the survey is suitable for completion by all stakeholders, items will be written in plain English with medical terminology and/or examples (where applicable) included in parentheses. The draft survey will be piloted by two professionals (eg, surgeons) and two members of the PPI group, to examine its face validity (eg, comprehensibility and acceptability) and refine terminology, layout and formatting prior to commencing the main study. Professionals will receive a version of the Delphi survey also including items related to reporting guidance content, which will be operationalised and formatted in a similar manner to outcomes.

### Participant sampling and invitations

Representatives from all key stakeholder groups detailed above will be invited to participate. Patients who are currently or who have previously participated in surgical

studies conducted by the Bristol Clinical Trials and Evaluation Unit will be invited to participate. Patients will be purposively sampled based on their gender, age, geographical region and study/procedure received to enable a broadly representative sample of participants with a range of experiences of different types of surgery to be included. Professional participants will be sampled to ensure inclusion of participants with various professional backgrounds and from broad geographical areas nationally and internationally. Professional participants will be identified through expert knowledge of colleagues (eg, known surgeons, specialty professional associations, industry collaborators and device manufacturers), contact lists held by the Centre for Surgical Research (University of Bristol) of attendees at topic-related academic events (eg, attendees at relevant conferences/workshops) and review of websites and other public resources. Additional patient and professional participants will be identified through opt-in to the Delphi survey via the COHESIVE study website, which will be further advertised through specialty professional associations described above.

### Delphi survey rounds

Participants will complete up to three sequential rounds of the survey questionnaire over a 3–9 months period. In each round, participants will score the importance of including the item in a COS (and reporting guidance—professionals only). Survey questionnaires will be predominantly administered online, hosted by secure REDCap electronic data capture software.[25] Administration and reporting of the electronic survey will be conducted in accordance with the Checklist for Reporting Results of Internet E-Surveys guidelines.[26] Paper surveys will be sent to those who are unable to complete the survey electronically.

All participants who complete the round 1 questionnaire will be sent the round 2 survey questionnaire. This second survey will contain all items retained from round 1 (see Data analyses section) and anonymised feedback from the previous round in the form of summary scores (eg, median/mean scores depending on distribution) from all stakeholder groups separately to allow all stakeholder groups to see the results from others before re-scoring; a method observed to improve the degree of consensus reached.[9 27] Participants will be asked to re-score each item's importance. If there is insufficient consensus to proceed to a consensus meeting at the end of round 2, a third round may be conducted. Methods will be identical to those described for round 2. All items retained after the final Delphi survey round will be taken forward to the consensus meeting.

### Methodological study exploring methods to optimise consensus

An embedded methodological study will explore optimal language for Delphi survey instructions. Participants will be randomly allocated to receive one of two versions of the round 1 survey questionnaire, which will differ only in the phrasing of the instructions provided for completing the questionnaire. Version A (standard instructions) will be phrased using common COS instruction terminology (eg, asking respondents to score how important they think each outcome/item is to include in the COS). Version B (enhanced instructions) will use terminology to emphasise that respondents should prioritise as few outcomes/items as possible. Participants will be allocated at random to one of the two groups and blinded to their allocation to minimise bias. Results from round 1 (eg, the number and types of outcomes that are prioritised) will be compared between the two allocation groups. In subsequent survey rounds, all participants will receive the same version of the survey questionnaire containing enhanced instructions to encourage prioritisation.

### Attrition between rounds

Participant attrition between rounds will be monitored and differences in scores between those who do and do not complete all survey rounds conducted.

### Phase III: consensus meeting with key stakeholders to discuss and agree the final reporting guidance content and COS(s)

The final step will be a face-to-face consensus meeting to discuss the results of the Delphi survey, agree on items that should be included, and approve the final guidance content and COS(s).[9 13] A purposive sample of 20–25 patient and professional participants who participated in at least one survey round will be invited, to ensure inclusion of patients with a range of experiences of different types of surgery and professionals with various professional backgrounds. A joint meeting for patients and professionals is planned to encourage discussion around outcomes that are scored differently by the two stakeholder groups, to share differing perspectives and thereby encourage consensus. Care will be taken to ensure that discussions are held using plain English language and that both patients and professional participants are empowered to express their views freely.

A summary of the survey results will be presented. Participants will be asked to ratify the inclusion/exclusion of outcomes during the Delphi survey, and to anonymously re-score items for which objections have been raised or for which consensus was not reached during the Delphi survey (see Data analyses section). Further moderated discussion and re-scoring will be undertaken as necessary until consensus has been reached. The meeting will be audio-recorded and transcribed verbatim. Another meeting will be considered should agreement on the final COS not be reached.

### Sample size

Delphi survey sample sizes aim to achieve good representation from all key stakeholder groups.[9] We aim to include approximately 150 professional participants and 50 patient participants in the survey and 20–25 participants from all stakeholder groups in the consensus meeting. In qualitative research, an approximation of sample size is

**Table 1** Definition of consensus

| Category | Definition | Action |
|---|---|---|
| Consensus in | Scored as very important (7–9) by ≥70% *and* not important (1–3) by <15% of either patients or professionals or both. | Item retained for next survey round/consensus meeting. |
| Consensus out | Scored as not important (1–3) by ≥70% *and* very important (7–9) by <15% of either patients or professionals or both. | Item discarded after round 2 (to be ratified at consensus meeting). |
| No consensus | Neither criteria above are met. | Item retained for next survey round/consensus meeting. |

considered necessary for planning. We anticipate that at least 25 interviews will be required to reach data saturation, though adequacy of the final sample size will be evaluated throughout the study. A 3:1 ratio of professionals to patient participants is considered appropriate given that there are multiple professional stakeholder subgroups whose involvement in the development of the COS is warranted.

### Data analyses
#### Retaining or dropping items between survey rounds
All data will be entered and stored on the REDCap electronic data capture tool[25] and exported into a statistical software package for data cleaning and checking. Descriptive statistics will be calculated for each item (eg, summary scores, ranges, percentage scoring each item 'not important' (score 1-3), 'equivocal' (4-6) and 'very important' (7–9)). All items will be retained between rounds 1 and 2.[9 28 29] This approach will enable participants to re-score every item considering feedback from round 1 but reduce participant burden in subsequent rounds (if necessary) and at the consensus meeting.[9] Following round 2, items will be categorised as described in table 1.

A third round will be held if the number of COS items categorised as 'consensus in' or 'no consensus' is considered too large (eg, >30) to feasibly discuss at the consensus meeting. The same criteria to define consensus and retain/discard items as above will be used. Stakeholder groups will be analysed separately in each round to negate the need to weight responses due to variation between sample sizes and to ensure that outcomes are not excluded prematurely. This methodology has been used in the development of other COS(s) to ensure that the outcome must be considered very important to the majority of participants to be included.[9 28–30]

#### Consensus meeting
Following the first round of voting, items will be categorised as 'consensus in', 'consensus out' or 'no consensus' using the definitions of consensus as detailed in table 1. Items voted 'consensus in' will be included in the final COS and items voted 'consensus out' discarded. Discordant items will be discussed further and re-scored in a second round of voting and the same criteria applied. Further rounds of discussion and voting will occur

until consensus is achieved. The consensus meeting will conclude with asking participants to ratify the final COS.

### Implementation of the COS(s)
Key multinational stakeholders will be engaged throughout the guidance/COS development process to promote awareness, foster ownership of the project and encourage dissemination and uptake. Engagement with funders, journal editors, regulatory bodies, industry and other bodies (eg, UK NIHR Surgical MedTech Cooperative, NIHR Office for Clinical Research Infrastructure) will seek to encourage and promote uptake in all studies of innovative invasive procedures/devices. We will also seek to involve these stakeholders and other international professional bodies responsible for promoting safe and transparent innovation (eg, in the UK: The Royal College of Surgeons of England, specialty professional associations, IDEAL) to endorse the reporting guidance and COS(s) and to encourage use in research and clinical practice. Patient and public engagement will establish the importance of this work more widely and encourage adoption of the guidance and COS(s) into other relevant guidelines/recommendations. The aim is to incorporate the reporting guidance and COS(s) as part of a real-time reporting system for surgical innovation. We will work with regulatory bodies and professional associations to develop a platform (eg, surgical registries that prospectively collect outcome and safety data) by which this may be achieved as the next steps of this work. Future work will also focus on how to measure the outcomes included in the COS(s).

This study was registered with the COMET Initiative on 20/11/2017.

### ETHICS AND DISSEMINATION
Ethical approval for this study has been granted by North East-Newcastle and North Tyneside 1 Health Research Authority Research Ethics Committee. Written informed consent will be obtained from participating patients separately for the Delphi survey and consensus meeting. Delphi survey registration via the study website and/or completion of the Delphi surveys by professional participants will be taken to imply consent. Written consent will

be obtained from professional participants prior to the consensus meeting.

Dissemination strategies include presentation at scientific meetings, peer-reviewed journal publications, development of plain English summaries and dissemination materials in collaboration with our PPI group, patient engagement events, development of a social media identity, workshops and other events.

**Author affiliations**
[1]Bristol Medical School, Population Health Sciences, University of Bristol, Bristol, UK
[2]Division of Surgery, University Hospitals Bristol NHS Foundation Trust, Bristol, UK
[3]Vascular Services, North Bristol NHS Trust, Bristol, UK
[4]Bristol Breast Care Centre, North Bristol NHS Trust, Bristol, UK

**Twitter** @CohesiveStudy

**Acknowledgements** The authors would like to acknowledge the role of the MRC HTMR funded workshop (Ref: N100; 01/2018-30/06/18; Principal Investigator: R Hinchliffe) and its attendees in informing the development of this study protocol. The authors would also like to acknowledge Sara Brookes for her input into the early conception and development of this study.

**Contributors** KA, JB and SP conceived and initiated the study and designed the protocol. KA and SP wrote the first draft of this manuscript. KA, JB, NSB, SC, DE, RH, RM, BM, SP, NW and JZ critically revised the protocol and manuscript.

**Funding** This study is supported by the National Institute for Health Research (NIHR) Biomedical Research Centre (BRC) at the University Hospitals Bristol NHS Foundation Trust and the University of Bristol. This work was supported by the Royal College of Surgeons of England Bristol Surgical Trials Centre and the MRC ConDuCT-II (Collaboration and innovation for Difficult and Complex randomised controlled Trials In Invasive procedures) Hub for Trials Methodology Research (MR/K025643/1) (www.bristol.ac.uk/social-community-medicine/centres/conduct2). The views expressed in this publication are those of the authors and do not necessarily reflect those of the UK National Health Service, the National Institute for Health Research, the Department of Health and Social Care, the Royal College of Surgeons of England or the Medical Research Council. SP is an NIHR Clinician Scientist (CS-2016-16-019). JB is an NIHR Senior Investigator. NSB and BM are NIHR Academic Clinical Lecturers.

**Competing interests** None declared.

**Patient consent for publication** Not required.

**Ethics approval** Ethical approval for this study has been granted by North East-Newcastle and North Tyneside 1 Health Research Authority Research Ethics Committee.

**Provenance and peer review** Not commissioned; externally peer reviewed.

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
