## [Reviewer comments · BMJ Open]

ARTICLE DETAILS

TITLE (PROVISIONAL)	Development of reporting guidance and core outcome sets for seamless, standardised evaluation of innovative surgical procedures and devices: a study protocol for content generation and a Delphi consensus process (COHESIVE study)
AUTHORS	Avery, Kerry; Blazeby, Jane; Wilson, Nicholas; Macefield, Rhiannon; Cousins, Sian; Main, Barry; Blencowe, Natalie; Zahra, Jesmond; Elliott, Daisy; Hinchliffe, Robert; Potter, Shelley

VERSION 1 – REVIEW

REVIEWER	Guy Maddern University of Adelaide Australia
REVIEW RETURNED	24-Mar-2019

GENERAL COMMENTS	An important project, put forward by an experienced group. My only concern is it remains very UK/European centric and I suggest engaging with more international agencies in Canada and Australia (eg CADTH, TGA), and attempting to include some surgeons from outside the UK in the process, beyond the Delphi program, who may have important non-NHS perspectives for device introduction into practice.
--

REVIEWER	Paul A. Stricker The Children's Hospital of Philadelphia and the Perelman School of Medicine at the University of Pennsylvania, USA
REVIEW RETURNED	14-May-2019

GENERAL COMMENTS	This protocol aims to fill an important void in providing formal, consensus-based guidance for outcomes selection to those involved in innovation. As the investigators note, this domain is outside the usual realm for which core outcome sets have been used and developed. The protocol is nonetheless best assessed in terms of its adherence to the fundamental principles of COS development as outlined in the COS-STAD recommendations. The scope and target audience are clear, the appropriate stakeholders will be involved, and the details of the consensus process are clear. The investigators have satisfactorily addressed each of the 11 key COS-STAD recommendations, with variations as appropriate for the niche subject of innovative procedures and devices.
--

VERSION 1 – AUTHOR RESPONSE

Reviewer 1:

1. My only concern is it remains very UK/European centric and I suggest engaging with more international agencies in Canada and Australia (eg CADTH, TGA), and attempting to include some surgeons from outside the UK in the process, beyond the Delphi program, who may have important non-NHS perspectives for device introduction into practice.

We thank the reviewer for this helpful suggestion. We agree that engaging a more diverse range of international stakeholders would be beneficial to maximise the representativeness of the reporting guidance and COS that will be developed and to promote their subsequent implementation and uptake. We have had international representation from the outset of the project but appreciate that we may not have made this clear in the submitted manuscript. For example, the workshop held convened in September 2018 to consider the feasibility of developing a COS/reporting guidance involved stakeholders representing several countries (including the UK, United States, Canada and Australia). We have revised the manuscript to add detail to this effect (detailed below). Since submitting the first draft of the manuscript, we have also pursued further international collaborations on the project. For example, we have included international representatives on the study steering group. We have also expanded our participant sampling strategy to increase the involvement of patient and professional stakeholder participants from a variety of countries, and now have expressions of interest from several international specialty patient groups and professional associations. In addition, we have expanded our plans to engage with international stakeholders and professional bodies to endorse the COS/reporting guidance and promote its implementation and uptake in research and clinical practice. We have revised the manuscript accordingly in the following places. All are minor revisions to the text:

- i. Page 2, line 22: words 'sampled from multi-national sources' added.
- ii. Page 3, line 8: word 'multi-national' added.
- iii. Page 5, line 4: revised to read '...multi-stage, international stakeholder Delphi survey'.
- iv. Page 6, line 20: new sentence added 'Stakeholders represented several countries, including the UK, United States, Canada and Australia'.
- v. Page 6, line 24: 'stakeholders' changed to 'stakeholder groups'.
- vi. Page 6, line 26: sentence changed to 'Primary stakeholders, recruited from international sources, will include:'.
- vii. Page 7, line 3: words 'from the UK and internationally' added.
- viii. Page 9, line 15: words 'nationally and internationally' added.
- ix. Page 9, line 17: words 'specialty professional associations' added.
- x. Page 9, line 21: words 'which will be further advertised through specialty professional associations described above.' added.
- xi. Page 10, line 15-16: words 'nationally and internationally' added.
- xii. Page 10, line 17: words 'specialty professional associations' added.
- xiii. Page 10, line 21-22: words 'which will further be advertised through specialty professional associations described above' added.
- xiv. Page 13, line 16: word 'multi-national' added.
- xv. Page 14, paragraph 1: the following sentences have been revised as follows: Engagement with funders, journal editors, regulatory bodies, industry and other bodies (e.g. UK NIHR Surgical MedTech Cooperative, NIHR Office for Clinical Research Infrastructure) will seek to encourage and promote uptake in all studies of innovative invasive procedures/devices. We will also seek to involve these stakeholders and other international professional bodies responsible for promoting safe and transparent innovation (e.g. in the UK: The Royal College of Surgeons of England, specialty professional associations, IDEAL) to endorse the reporting guidance and COS(s) and to encourage use in research and clinical practice.

Other minor revisions

We have made a small number of minor revisions to the text for accuracy or to correct errors:

- i. Page 1: first names of three authors corrected.
- ii. Page 2, line 23: words 'minimum two rounds' removed due to word limit.
- iii. Page 2, line 29: text revised to read 'scientific meeting presentations' due to word limit.
- iv. Page 3, line 20: word 'or' changed to 'nor'.
- v. Page 6, line 4: word 'emerging' replaced with 'ongoing' and related reference added.
- vi. Page 8, line 10: we have revised the text describing the targeted review of studies self-identifying as IDEAL, to reflect the fact we are now reviewing papers that cite more of the original IDEAL publications for this aspect of the study. The sentence now reads "Studies citing the three original 2009 Lancet publications[6, 16, 17], subsequent explanatory documents[18-23] or the IDEAL-D[7] paper...". The reference list has also been updated accordingly.
- vii. Page 9, line 24: we have revised the text to give detail about how the Likert scale in the Delphi survey questionnaire will be formatted. The sentence now reads 'Each item will have a 9-point Likert scoring scale ranging from 1 (not important) to 9 (extremely important) based on the Grading of Recommendations Assessment, Development and Evaluation (GRADE) scale for scoring the importance of including the item in the final COS[23]'.
- viii. Page 10, line 2: we have revised the text to describe that survey questionnaire items may include examples as follows: 'items will be written in plain English with medical terminology and/or examples (where applicable) included in parentheses.'
- ix. Page 10, line 9: word 'six' removed.
- x. Page 10, line 27: sentence 'Administration and reporting of the electronic survey will be conducted in accordance with the Checklist for Reporting Results of Internet E-Surveys (CHERRIES) guidelines' added.
- xi. Page 13, Box 1: word 'critically' changed to 'very' (two instances).
- xii. Page 13, line 1: word 'COS' added.
- xiii. Page 14: we have updated the Author Contributions section to ensure all initials/names of authors are individually listed.
- xiv. Page 15: we have updated the funding statement to reflect recent changes in the way funders request their support to be stated, and to include details of an additional author grant.